# Accumulation *γ-*Aminobutyric Acid and Biogenic Amines in a Traditional Raw Milk Ewe’s Cheese

**DOI:** 10.3390/foods8090401

**Published:** 2019-09-10

**Authors:** Rosanna Tofalo, Giorgia Perpetuini, Noemi Battistelli, Alessia Pepe, Andrea Ianni, Giuseppe Martino, Giovanna Suzzi

**Affiliations:** Faculty of BioScience and Technology for Food, Agriculture and Environment, University of Teramo, 64100 Teramo, Italy

**Keywords:** *γ*-aminobutyric acid, biogenic amines, raw milk ewe’s cheese, Pecorino di Farindola, histidine decarboxylase (*hdc*) gene, tyrosine decarboxylase (*tdc*) gene

## Abstract

The influence of calf (R1), kid (R2) and pig (R3) rennets on microbiota, biogenic amines (BAs) and *γ*-aminobutyric acid (GABA) accumulation in raw milk ewe’s cheeses was evaluated. Cheeses were investigated at different ripening times for their microbial composition, free amino acids (FAAs), BAs and GABA content. Moreover, the expression of tyrosine (*tdc*) and histidine (*hdc*) decarboxylases genes was evaluated by quantitative Real Time–Polymerase Chain Reaction (qRT-PCR). Microbial counts showed similar values in all samples. Pig rennet were cheeses were characterized by higher proteolysis and the highest values of BAs. The BAs detected were putrescine, cadaverine and tyramine, while histamine was absent. qRT-PCR confirmed this data, in fact *hdc* gene was not upregulated, while *tdc* gene expression increased over time in agreement with the increasing content of tyramine and the highest fold changes were detected in R3 cheeses. GABA showed the highest concentration in R2 cheeses reaching a value of 672 mg/kg. These results showed that the accumulation of BAs and GABA in Pecorino di Farindola is influenced by ripening time and type of coagulant. Further studies are required to develop starter cultures to reduce BAs content and improve health characteristics of raw milk ewe’s cheeses.

## 1. Introduction

Pecorino di Farindola is an artisanal cheese of the Abruzzo region (Italy) produced following traditional practices. Farindola is a town located in National Park of Gran Sasso, Italy, at an altitude of 530 m (1740 ft). This cheese is produced only in this geographical area and has a soft texture with a thin yellow rind or can appear with a harder texture and intense/piquant flavour as the ripening time increases. It is exclusively produced with raw ewes’ milk and pig rennet without the use of natural or commercial starter cultures [1,2,3]. 

Lamb and kid rennet are the main coagulants used in Mediterranean countries for the production of Protected Denomination Origin (PDO) ovine and goat cheeses, like Pecorino Romano, Fiore Sardo and Canestrato Pugliese in Italy and Feta cheese in Greece [4,5]. Di Giacomo et al. [6] reported that pig rennet was already used by the ancient Romans to produce a “cheese of Vestini” (an ancient tribe of Abruzzo). Pig rennet is obtained from stomach mucus membrane and after an incubation of 2–3 days in salt, it is mixed with white vinegar, white wine and chili pepper and stored for 3–4 months. Finally, it is filtered for 5–6 days and only at this point it is ready to use [7,8,9]. Pig pepsin is unstable above pH 6.0 [9]. Its clotting activity strongly depends on pH, in fact coagulation does not occur above pH 6.7 [10]. Previous studies evaluated the influence of pig rennet in the manufacture of Pecorino di Farindola in terms of physico chemical properties, microbiota, proteolysis, volatile molecule profiles and other characteristics [3,9]. The studies showed that the use of pig makes it possible to distinguish the traditional variant from cheeses made with other coagulants.

Pecorino di Farindola cheese production is not standardized and autochthonous lactic acid bacteria (LAB) are the main responsible of the definition of final product characteristics [1]. On the other hand, non-starter lactic acid bacteria (LAB) deriving from raw milk or from the dairy environment play an important action during ripening in terms of sensory characteristics and safety issues, such as biogenic amines (BAs) accumulation [11]. Type of cheese, ripening time, manufacturing process and microorganisms highly influence the BAs content [12] and for this it can be extremely variable. BAs were found in Pecorino di Farindola cheeses examined by Schirone et al., [1]. Their total content ranged from 209.0 to 2393.0 mg/kg cheese and tyramine was the main BA detected. 

BAs, mainly histamine, can lead to intoxications and adverse reactions to human health, especially after the ingestion of food products rich in BAs content [13,14]. The “cheese syndrome” caused by tyramine is rather a side effect of monoamino oxidase (MAO) drugs than a food safety issue. The consumption of cheese containing tyramine is unlikely to cause health problems in healthy individuals. EFSA report showed the absence of negative effects on health for healthy individuals after the exposure to following BA levels in food: (a) 50 mg histamine; (b) 600 mg tyramine for individuals not taking monoamino oxidase inhibitor (MAOI) drugs, but 50 mg for those using third generation MAOI drugs or 6 mg for those taking classical MAOI drugs. For putrescine and cadaverine not enough information is available [15].The intoxications related to BAs consumption can get worse in association with alcohol, other amines and monoamine and diamine oxidase-inhibiting drugs, resulting in serious problems for human health [14]. Diamine oxidase inhibitor (DAOI) drugs can also be responsible of histamine related symptomatology. Monoamine oxidase (MAO), diamine oxidase (DAO) and polyamine oxidase (PAO) are enzymes naturally present in the organism and responsible of BAs detoxification through acetylation and oxidation [16,17]. 

Moreover, in cheeses, there are also substances present without a defined nutritional function which could have a beneficial impact on human health and, among these compounds, *γ*-aminobutyric acid (GABA) has been reported with numerous positive effects on animal and human metabolic disorders [18]. This compound is synthesized by glutamate decarboxylase (GAD) catalysing the decarboxylation of L-glutamate to GABA [19]. Some studies revealed the ability of this to decrease arterial pressure and to reduce blood pressure in hypertensive patients [20,21,22]. Therefore, several GABA-enriched food products have been manufactured, such as GABA-enriched green tea [23], rice germ [24], tempeh and fermented beverages [25]. Dairy products fermented with GABA-producing LAB have also been studied and found to have physiological effects [26,27]. In fact, native caseins contain a high proportion of L-glutamate that can be released during milk fermentation and proteolysis. 

Because of its positive impact on health, it gained the attention of the food and pharmaceutical industries. GABA is a non-protein amino acid with multiple physiological functions produced by some yeasts and bacteria. Recently, the development of functional GABA-enriched foods, such as cheese, have been reported [26,27]. In this study the role of pig rennet in the safety hazards and bioactive compounds production in Pecorino di Farindola was evaluated. In particular, cheeses were produced with three different coagulants (pig, lamb and kid) and obtained cheeses were compared in terms of microbiota, BAs and GABA. 

## 2. Materials and Methods 

### 2.1. Cheese Manufacture

Cheese samples were made with raw ewe’s milk according to the traditional protocol [23]. Calf (R1), kid (R2) (Colombo s.r.l., Sirtori, Italy) and pig (R3) rennets (Azienda Agricola Martinelli Pietropaolo, Farindola, Italy), were used to coagulate three different milk batches obtained during the milking day, as reported by Tofalo et al. [9]. Three experimental batches consisting of 24 cheeses were carried out. After pressing and dry salting, the cheeses (∼2.8 kg each) were placed in a ripening chamber at 14–15 °C. Analysis were performed after 7, 15, 30, 60, 90, 180 and 270 days of ripening.

### 2.2. Microbial Analysis

For the microbiological analysis, serial dilutions in sodium citrate (2% *w/v*) were prepared starting from 10 g of each cheese The following microorganisms were investigated: aerobic mesophilic bacteria (AMB) on Plate Count Agar (PCA; Oxoid, Milan, Italy) at 30 °C for 2 days; mesophilic lactobacilli on MRS agar (Oxoid, Basingstoke, UK), acidified to pH 5.4 with acetic acid, at 30 °C for 2 days in anaerobic conditions using the Gas-Pack anaerobic system (AnaeroGen; Oxoid, Basingstoke, UK); lactococci on M17 (Oxoid, Basingstoke, UK), containing 1% (*w/v*) lactose (Fluka Chimica, Milan, Italy), at 30 °C for 2 days in anaerobic conditions; enterococci on Slanetz-Bartley agar (Oxoid, Basingstoke, UK) at 37 °C for 48 h; *Enterobacteriaceae* on Violet Red Bile Glucose Agar (VRBGA; Oxoid, Basingstoke, UK) at 37 °C for 24 h. Cell counts were performed in duplicate.

### 2.3. qPCR Analysis

Tyrosine decarboxylase (*tdc*) and histidine decarboxylase (*hdc*) genes were detected as described by Nadkarni et al. [28], Torriani et al. [29] and Fernández et al. [30]. Primer pairs and qPCR conditions are reported in Table 1. Total RNA was extracted using a MO BIO RNA Power Soil Kit (QIAgen, Milan, Italy), according to the manufacturer’s instructions. The possible presence of contaminating DNA was checked by PCR and eventually the DNase treatment was repeated. One μg of total RNA was retrotranscribed using the iScript™ cDNA Synthesis Kit (Bio-Rad, Milan, Italy), according to the manufacturer’s instructions. Real-time analysis was performed using an iCycler IQ realtime PCR Detection System (Bio-Rad, Milan, Italy). A reaction mixture of 25 μL containing 12.5 μL 2XIQ SYBR Green PCR Supermix™ (Bio-Rad, Milan, Italy), 0.2 μmol/L of each primer (Life Technologies-Invitrogen, Milan, Italy) was prepared. Fold changes were determined as previously described [31]. 16S rRNA was used as reference genes. Its relative stability was evaluated using NormFinder program [32]. After real-time PCR, a melting-curve analysis was performed by measuring fluorescence during heating from 50 to 95 °C at a transition rate of 0.2 °C/s to verify the presence of unspecific products or primer dimers. A single peak was obtained highlighting the specificity of the amplification. All analyses were performed in triplicate.

### 2.4. Free amino Acids (FAAs)

Free amino acids (FAAs, expressed as mg leucine/g) were evaluated at 507 nm after reaction with Cd-ninhydrin according to Folkertsma and Fox [33]. Analyses were performed in triplicate on each sample.

### 2.5. Biogenic Amines Determination 

Determination of BAs was performed by acid extraction and derivatization according to Eerola, et al. (1993) [34], and Moret and Conte [35], as reported by Tittarelli et al. [17]. The presence of putrescine, cadaverine, tyramine, histamine, spermidine and spermine was determined homogenizing 2 g of cheese in 20 mL of 0.1 M HCl containing 100 mg/L of 1,7-diaminoheptane (Fluka, Milano, Italy) used as internal standard. A Waters Alliance High Performance Liquid Cromatography (HPLC) system (Waters SpA, Vimodrone, Italy), equipped with a Waters 2695 separation module connected to a Waters 2996 photodiode array detector was used. Analytes were separated using a Waters Spherisorb C18 S3ODS-2 column (3 μm particle size, 150 mm × 4.6 mm I.D.). Acetonitrile (A) and ultrapure water (B) were used for the separation of BAs. The following elution gradient was applied: 57% A for 5 min; concentration was increased up to 80% linearly in 4 min, 90% A for 5 min. The flow rate was 0.8 mL/min and the column temperature was set at 30 °C ± 0.1 °C. The peaks were detected at 254 nm. The system was controlled by Waters Empower personal computer software. Identification of the BAs was based on their retention times. 

### 2.6. GABA Determination

GABA was determined according to Kőrös, et al. [36] and Ianni et al. [37]. Analyses were performed using the High Pressure chromatographic system (Perkin-Elmer, Monza, Italy) equipped with an autosampler and a UV-VIS detector set at 210 nm with an ion-exchange column (Phenomenex Gemini C18, dimensions: 250 × 4.6 mm, particle size: 5 μm, pore size: 110 Å) (Phenomenex, Bologna, Italy). GABA was derivatized with phthaldialdehyde Reagent (Sigma-Aldrich, Milan, Italy) and injected into the HPLC system. 

## 3. Results and Discussion

Due to the increasing awareness towards the impact of diet on human health, issues relating to food safety and quality, have a crucial role on the consumers’ behaviour. Therefore, dairy industries are developing foods with improved nutritional quality. Cheese is one of the most important fermented food product. Despite it is rich in positive compounds for human health (e.g., GABA) [38], it is also associated to BAs intoxication [13]. For this reason, the European Food Safety Authority (EFSA) Panel on Biological Hazards (BIOHAZ) has put forth an opinion on risk assessment related to BAs pointing out that the actual knowledge of their toxicity is still limited, and that further research is needed [15]. In this study, the influence of different animal rennets on microbiota, BAs and GABA accumulation in Pecorino di Farindola cheese during ripening was evaluated. 

### 3.1. Microbial Analysis

The evolution of microbial populations at different days of ripening is reported in Figure 1. In general, no significant differences were observed. After 90 days of ripening a decrease of cell counts was detected for all microbial groups, with *Enterobacteriaceae* disappearing. The highest values of *Enterobacteriaceae* were detected in R2 cheeses with cell counts of 3.69 Log CFU/g at 90 days. 

*Enterobacteriaceae* are an indicator of the hygienic conditions in milk and cheese production. Their occurrence has been reported in some raw milk cheeses of the Mediterranean basin after 30 days of ripening [39,40]. Moreover, some studies highlighted that this microbial group can influence taste, aroma and texture, of some artisanal cheeses [41,42]. Enterococci were present with cell counts of about 6 Log CFU/mL in all samples until 90 days of ripening, while after this time a decrease of about 3 Log was observed. Their occurrence has been reported in other Pecorino cheeses [9,43]. Their presence can be due to milk contamination and to their ability to face the conditions of cheese manufacture and ripening since they are able to develop at different temperatures and are tolerant to heat and salt [44]. Moreover, they have a crucial role in cheese ripening and aging influencing aroma and flavour thanks to their proteolytic and lipolytic activities as a result of citrate metabolism [44]. On the other hand, enterococci have often been associated with clinical infections and BAs production, such as tyramine and histamine [45]. 

Lactococci were present in lower concentration at the end of ripening, about 4 Log CFU/mL in R2 and R3 cheeses and about 2 Log CFU/mL in R1 cheese. LAB counts throughout the ripening were similar to those observed for AMB suggesting that they were the predominant microorganisms. Similar results have been reported by Renes et al. [46]. Their counts increased during the first days of ripening reaching values of about 8 Log CFU/g in R2 and R3 cheeses after 90 days of ripening, while in R1 cheeses cell counts of about 7 Log CFU/mL were observed. LAB are the main components of the autochthonous cheese and are known to participate to the fermentation process and maturation of cheeses, producing a number of desirable substances that can improve the flavour, texture, nutritional value, shelf-life, and safety of foods [46,47,48]. However, some LAB species have been shown to include strains producing high amounts of BAs such as histamine and tyramine [46,49,50,51,52,53,54]. 

### 3.2. Biogenic Amines 

BAs accumulation in cheese depends on several factors including ripening time, the manufacturing process, presence of decarboxylase positive microorganisms and precursors availability [14,46]. In general, during cheese ripening secondary proteolysis occurs with the accumulation of FAAs which can be decarboxylated to BAs by the microbiota [46]. Therefore, the level of protein degradation was firstly evaluated. 

Figure 2 shows the evolution of FAAs during the ripening period. Obtained data revealed a certain variability of FAAs content among samples. The FAAs content increased significantly during ripening in all samples, reaching values of 233.74, 252.54 and 296.84 mg leucine/g in R1, R2 and R3 cheeses respectively after 270 days of ripening. This increase is in agreement with previous observations in Caciocavallo Pugliese, in Picante cheese, in Kashkaval cheese and in other Pecorino cheeses [9,55,56,57,58]. The greater proteolytic activity in R3 cheeses was in agreement with Tofalo et al., (2015) [9] who observed a higher proteolytic activity for pig rennet than for calf and kid ones. R3 cheeses contained the highest total concentrations of BAs (1293 mg/kg) (Figure 3). However, all the studied cheeses accumulated high total BAs contents even if with quantitative differences. It could be assumed that milk quality produced by the sheep in the Gran Sasso area is a relevant factor influencing the high BAs content of Pecorino di Farindola, probably also depending on the autochthonous microbiota [59,60]. 

The main BAs detected were putrescine, cadaverine and tyramine, while histamine was not detected. Putrescine showed the highest concentration with values of 400 mg/kg, 598 mg/kg and 732 mg/kg in R1, R2 and R3 cheeses, respectively after 270 days. Cadaverine presented values ranging from 241 mg/kg (R1) to 154 mg/kg (R2) at the end of ripening. These BAs are considered indicators contamination and markers of the hygiene standards of the production process. Their production mainly relies on Gram-negative bacteria, especially belonging to the families *Enterobacteriaceae* and *Pseudomonadaceae*, generally associated with spoilage [61]. LAB and staphylococci have also been reported [62]. Tyramine was present in all samples and its concentration increased over time. To our knowledge, Enterococci play a predominant role in the formation of tyramine [63]. The high BAs content detected in cheeses could be related to the productions practices associated to Pecorino di Farindola. In fact, it is produced starting from raw milk from sheep fed in a limited mountain area (Gran Sasso) and pig rennet as coagulant. The combination of these two factors could be a cause of high BAs content in the cheese. In fact, it has been already proved that there is a relationship between alpine pastures and milk quality during grazing [64].

Tyramine presents several negative effects on human health such as headaches, migraine, neurological disorders, nausea, vomiting, respiratory disorders, hypertension [13,65]. However, even though histamine was considered as the most toxic BA for a long time, recently Linares et al. [66] revealed that tyramine is even more toxic than histamine reporting the cytotoxicity threshold detected for histamine (441 mg/kg) and for tyramine (302 mg/kg). Amino acid decarboxylase activity is strain dependent rather than species specific, thus suggesting the occurrence of horizontal gene transfer events as part of a mechanism of survival and adaptation to specific environments [67]. Wüthrich et al. [68] sequenced the histamine positive strain FAM21731 of *L. parabuchneri* showing that *hdc* gene cluster was located in a genomic island, transferred within this species. This species has been frequently reported in dairy products and is responsible for the accumulation of histamine in many types of cheeses [49,50,51]. Its occurrence in milk is probably related to a contamination focus at farm level, because its capacity to adhere to stainless steel [50,52,53]. Moreover, *L. parabuchneri* strains are able to develop and to produce histamine also at refrigeration temperatures, suggesting that when *L. parabuchneri* is present, refrigeration can only delay but not prevent the accumulation of histamine in cheese [54]. For this reason, aminogenic strains may be found within the contaminant species but also as part of the spontaneous fermentative microbiota. Therefore, the expression of *tdc* and *hdc* genes was checked by qRT-PCR. In all samples *hdc* gene was not upregulated confirming the absence of histamine in cheeses, whereas *tdc* expression increased over time in agreement with the increasing content of tyramine (Figure 4). The strongest increase in *tdc* expression was found in R3 cheese after 270 days of ripening (a 93-fold increase), whereas the other cheeses showed 61- (R1) and 82- (R2) fold increases of the *tdc* gene. 

This development was expected, because in LAB, BAs formation provides metabolic energy and/or acid resistance during a long ripening [69,70]. In general, a positive correlation between *tdc* expression and tyramine production was observed (Figure 4). This evidence is in agreement with other studies and suggests that the expression of this gene can be used to predict tyramine accumulation [13,17,43]. The high content of tyramine could be related with the high content of putrescine. In fact, agmatine deiminase pathway genes, involved in putrescine production, are linked to the tyrosine decarboxylation operon in LAB [71]. The agmatine deiminase and the tyrosine decarboxylase pathways, appear to be widespread throughout several species of LAB and are often simultaneously present [72]. Moreover, the formation of the different BAs could be also influenced by the microbial contamination in the processed milk. 

### 3.3. γ-Aminobutyric Acid

The GABA content determined in this study is reported in Figure 5. The highest concentrations of GABA were found in R2 and R3 cheeses, with values of 672 and 554 mg/kg, respectively. 

These results are in agreement with the proteolysis outcome, in fact in R1 cheeses a slower proteolysis was observed. In general, GABA production is associated to different LAB species in a strain dependent way [73]. Siragusa et al. [73] studied 22 Italian cheese varieties for their GABA concentrations that varied from 0.26 to 391 mg/kg. The highest values of GABA were found in five varieties of Pecorino cheeses, but especially in Pecorino di Filiano (391 mg/kg). Moreover, Nomura et al. [74] analysed seven commercial cheeses (Camembert, Gouda, Blue, Cream, Cheddar, Edam and Emmental) and reported GABA concentration lower that those obtained in this study. In general ewe’s milk quality and coagulant seem to have some influence on GABA accumulation in cheese.

## 4. Conclusions

Traditional cheeses like Pecorino di Farindola, maintain high diversity in cheese-making practices as well as in autochthonous cheese microbial communities. The combination of hygienic quality of raw ewe’s milk, the different handling and cheesemaking processes and pig rennet could be a cause of its high BAs content. However, it is expected to see a total BAs content rise in cheese as a consequence of many BAs precursors released by coagulants from a raw milk with a high proteins content. Pig rennet imparts specific features to Pecorino di Farindola cheese, probably through its specific proteolytic patterns resulting from the unique enzymatic composition of this coagulant. However, this study confirms a greater BAs formation and proteolytic activity, as suggested by FAA values, in cheese made by pig rennet than those made by calf and kid rennet. It is difficult to modify the process without denaturing the organoleptic and sensorial characteristics of this traditional cheese. A possibility could be the pasteurization of raw milk, but this procedure denatures milk enzymes and reduces the levels of milk natural microbiota. Moreover, the pasteurized cheeses are negatively perceived by consumers compared with the sensory characteristics of cheese made with raw milk. A possible strategy for reducing BAs accumulation, increasing the safety and maintaining the sensorial characteristics of traditional cheese could be the selection of autochthonous amine-negative and amine-oxidizing LAB. These strains could be used as a starter or an adjunct/attenuated starter. In addition, Pecorino di Farindola contains high amounts of GABA which can be correlated with the use of ewe’s milk, time of ripening and type of coagulant. This cheese together with other Italian “Pecorino” cheeses could be a source of microorganisms able to synthesize GABA for the production of dairy products with functional and probiotic properties.

## Figures and Tables

**Figure 1 foods-08-00401-f001:**
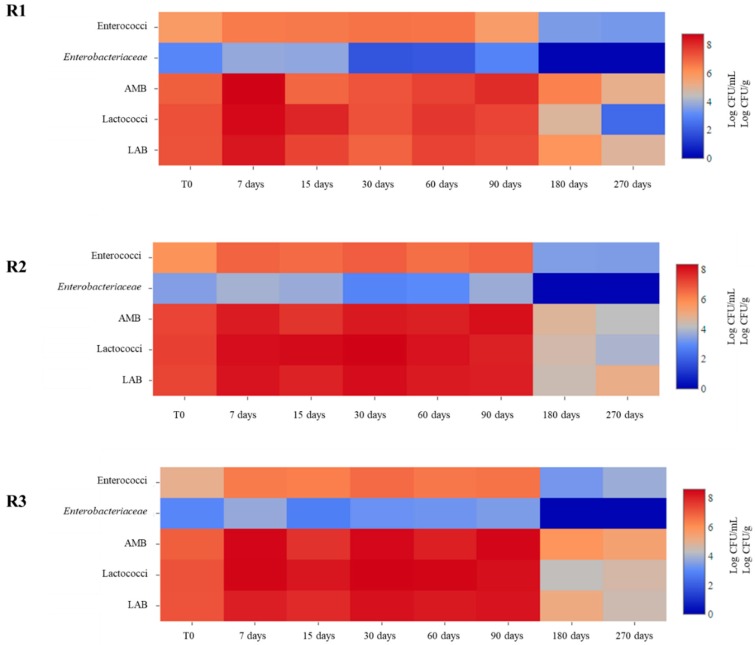
Heat maps depicting abundance of microbial groups characterizing the cheeses obtained with different rennets throughout ripening. Calf (**R1**), kid (**R2**) and pig (**R3**) rennets. Aerobic Mesophilic Bacteria (AMB); Lactic Acid Bacteria (LAB).

**Figure 2 foods-08-00401-f002:**
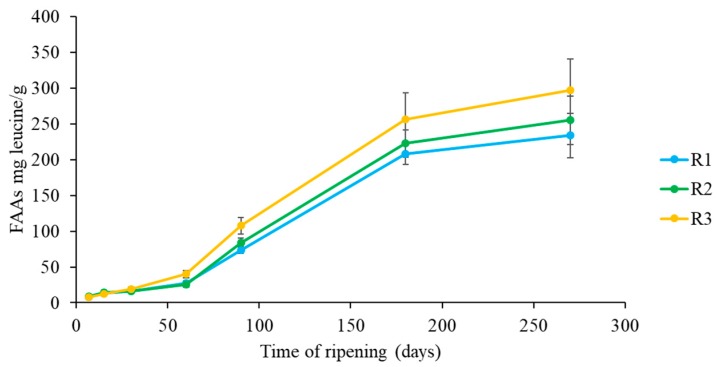
Evolution of free amino acids throughout ripening in Pecorino di Farindola produced with with different rennets. calf (R1), kid (R2) and pig (R3).

**Figure 3 foods-08-00401-f003:**
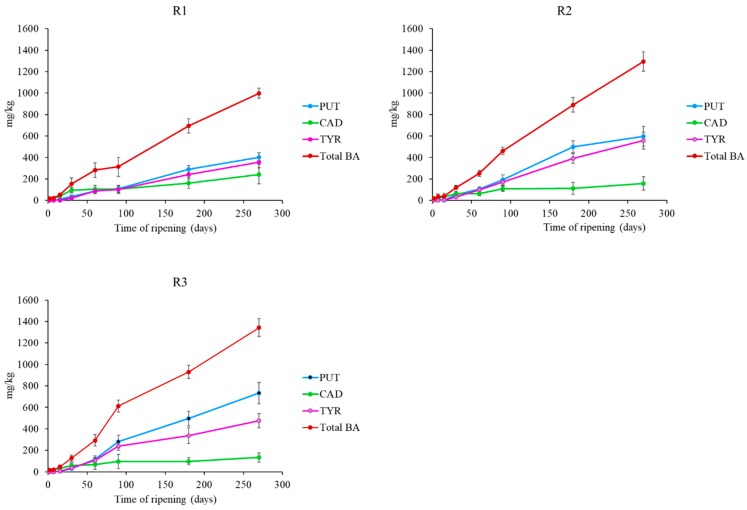
Biogenic amines evolution in Pecorino di Farindola manufactured with the three different rennets: calf (R1), 216 kid (R2) and pig (R3). Putrescine (PUT); Cadaverine (CAD); Tyramine (TYR); Biogenic Amine (BA).

**Figure 4 foods-08-00401-f004:**
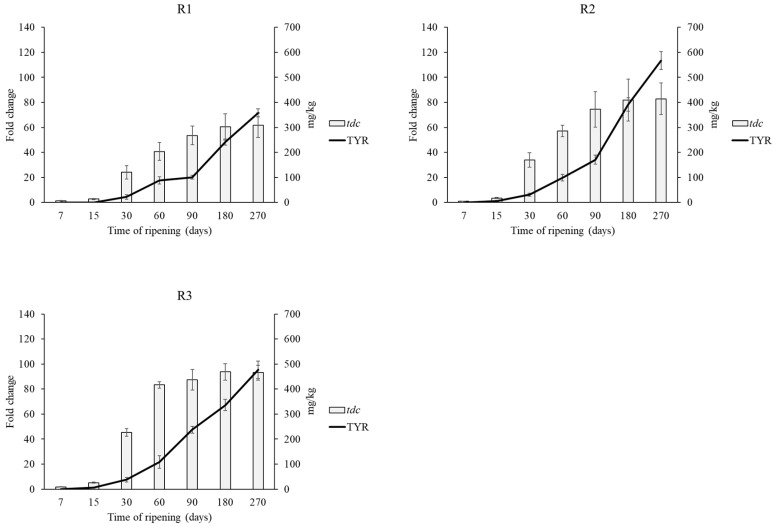
Tyramine content and relative transcript levels of *tdc* gene during ripening. Transcript levels are expressed as x-fold increase in comparison with the expression at T0. Three biological replicates were performed. Calf (R1), kid (R2) and pig (R3).

**Figure 5 foods-08-00401-f005:**
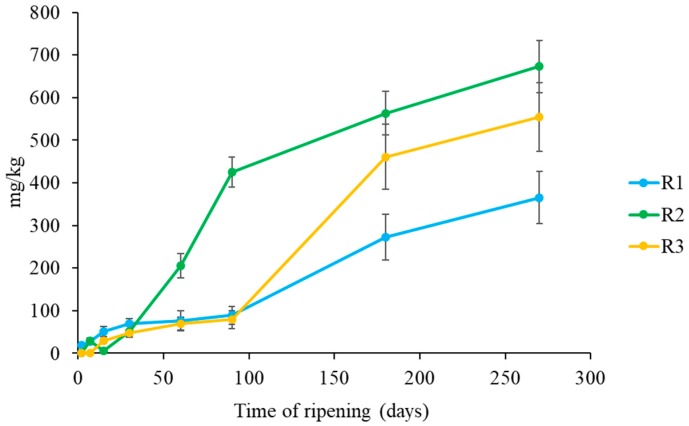
Evolution of GABA content in Pecorino di Farindola made with the three different rennets: calf (R1), kid (R2) and pig (R3).

**Table 1 foods-08-00401-t001:** Primer sequences and PCR conditions used in this study.

Primer	Sequence (5′-3′)	qPCR Conditions	References
***16SF*** ***16SR***	TCCTACGGGAGGCAGCAGT GGACTACCAGGGTATCTAATCCTGTT	95 °C for 10 min, 40 cycles at 95 °C for 15 s, 60 °C for 1 min, 72 °C for 45 s	[28]
***Tyr3*** ***Tyr4***	CGTACACATTCAGTTGCATGGCAT ATGTCCTACTTCTTCTTCCATTTG	94 °C for 5 min, 35 cycles at 94 °C for 20 s, 58 °C for 30 s, 72 °C for 45 s	[29]
***Hdc1*** ***Hdc2***	TTGACCGTATCTCAGTGAGTCCAT ACGGTCATACGAAACAATACCATC	95 °C for 10 min, 40 cycles at 95 °C for 15 s, 58 °C for 1 min, 72 °C for 45 s	[30]

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
