# Peer review of "Accumulation γ-Aminobutyric Acid and Biogenic Amines in a Traditional Raw Milk Ewe’s Cheese"

_foods, 2019, doi:10.3390/foods8090401_

Round 1

Reviewer 1 Report

Peer_Review_Foods-582135_v1
Title: Accumulation -aminobutyric acid and biogenic amines in a traditional raw milk ewe’s cheese,
General comments:
The present manuscript describes a study on the formation of BAs and GABA in Pecorino di Farindola that is presumably the only cheese variety in the world prepared a traditional preparation of pork rennet.
The experimental design is sound, the type of performed analysis comprehensive. The results are well presented. However, the discussion of the results and mainly the conclusions should be refined. A more comprehensive view of the factors leading to the formation of BA is needed in order to embed the obtained results in the right context.
Without doubt, the manuscript is well written and contains substantial information and therefore should be after major revision.
Specific comments:
Abstract:
Lines 20-23: These results showed that the accumulation of BAs and GABA in Pecorino di Farindola is influenced by ripening time and type of coagulant. Further studies 21 are required to develop starter cultures to reduce BAs content and improve health characteristics of raw milk 22 ewe’s cheeses.
Introduction:
Line 42: Please insert some information on the characteristics of pig rennet in comparison to other rennet types (e.g. proteolytic activity, pH optimum, thermo-resistance).
Line 44: Please consider to rephrase so then the next sentence on NSLAB is better embedded in the context. The microbial composition of the lactic acid bacteria (LAB) present in the starter plays a major role…..On the other hand, adventitious…”
Line 55: Please specify “Biogenic amines, mainly histamine, can lead to …”. The "cheese syndrome" caused by tyramine is rather a side effect of MAO drugs than a food safety issue. The consumption of cheese containing tyramine is unlikely to cause health problems in healthy individuals. According to the EFSA report no adverse health effects were observed after exposure to following BA levels in food (per person per meal): a) 50 mg histamine for healthy individuals, but below detectable limits for those with histamine intolerance; b) 600 mg tyramine for healthy individuals not taking monoamino oxidase inhibitor (MAOI) drugs, but 50 mg for those taking third generation MAOI drugs or 6 mg for those taking classical MAOI drugs; and c) for putrescine and cadaverine, the information was insufficient in that respect.for (EFSA Journal 2011;9(10):2393).
Line 64: Please add a short listing of the most important positive effects claimed for GABA. … numerous positive effects numerous positive effects on animal and
human metabolic disorders such as ……………[16] and therefore gained the
attention of the food and pharmaceutical industries.
Lines 68-73: Move these sentences to line 34 and consider rephrasing
….cultures [2,3]. In previous studies, the role of pig rennet in the manufacture
of Pecorino di Farindola on cheese quality was evaluated. The effect of
different types of rennet of animal origin (calf, kid and pig) was compared in
terms of physico chemical properties, microbiota, proteolysis, volatile molecule
profiles and other characteristics [3,9]. The studies showed that the use of pig
makes it possible to distinguish the traditional variant from cheeses made with
other coagulants.
Materials and Methods:
Line 92: nsert a semicolon (;) …….conditions; enterococci……
Line 98: Replace the semicolon by a comma …….[15], Torriani ………..
Line 113: Delete the comma after Kőrös …Kőrös et al.
Line 116: Please indicate the column type as follows: (Gemini C18, dimensions: 250 ×
4.6 mm, particle size: 5 μm, pore size: 110 Å) (Phenomenex)).
Results:
Line 117: “Results and discussion” would be a more appropriate title
Line 132: The milking process is important as well. Consider to modify as follows: ….an
indicator of the hygienic conditions in milk and cheese production.…..
Line 132-133 Delete this part of the sentence …and their absence in the final products
reveals good hygienic conditions during the manufacturing process…
(Enterobacteriaceae may serve as a process hygiene criterion in cheeses just
after manufacture (day 1-14). However, as observed, Enterobacteriaceae
disappear rather quickly during cheese ripening. Therefore, the absence of
enterobacteriaceae in ripened cheese is not a reliable indicator for good
hygienic conditions during the manufacturing process.)
Line 143: Change this sentence as follows: On the other hand, enterococci have often
been associated with clinical infections and BAs production, such as tyramine
and histamine.
Line 150: It's the other way around. High levels of FAA are rather due to specific species
of LAB such as L. helveticus. Consider to delete this sentence
Line 154: Reference 35 focuses on -aminobutyric acid and should be cited together
with reference 36 & 37….[35-37]. However,……..
Line 154: Recently, Lactobacillus parabuchneri has been shown to be the most potent
LAB species for the formation of histamine. Consider to rephrase as follows:
However, some LAB species have been shown to include strains producing
high amounts of BAs such as histamine [new references].
Please consider to include these references in order to update the discussion.
Diaz M., Ladero V., Redruello B., Sanchez-Llana E., del Rio B., Fernandez B., Cruz
Martin M., Alvarez M.A. (2016). A PCR-DGGE method for the identification of
histamine-producing bacteria in cheese. Food Control 63 (2016) 216e223
http://dx.doi.org/10.1016/j.foodcont.2015.11.035
Berthoud H. Wüthrich D., Bruggmann R., Wechsler D., Fröhlich-Wyder M.-T., Irmler
S. (2017). Development of new methods for the quantitative detection and typing of
Lactobacillus parabuchneri in dairy products. International Dairy Journal, 70, 65-71.
http://dx.doi.org/10.1016/j.idairyj.2016.10.005
Ascone P., Maurer J. Haldemann J., Irmler S., Berthoud H., Portmann R., Fröhlich-
Wyder M.-T., Wechsler D. (2017). Prevalence and diversity of histamine-forming
Lactobacillus parabuchneri strains in raw milk and cheese - a case study. International
Dairy Journal, 70, 26-33.
http://dx.doi.org/10.1016/j.idairyj.2016.11.012
Diaz M., del Rio B., Sanchez-Llana E., Ladero V., Redruello B., Fernández M., Cruz
Martin M., Alvarez M.A. (2018). Lactobacillus parabuchneri produces histamine in
refrigerated cheese at a temperature-dependent rate. International Journal of Food
Science and Technology, 53,2342–2348
http://dx.doi.org/10.1111/ijfs.13826
Line 172: As expected, … Long ripened high quality raw milk cheeses such as
Parmigiano Reggiano PDO or Emmental PDO usually contain up to 40 g/kg
FAA but < 50 mg/kg BA., indicating that there is no correlation between the
FAA content and the concentration of BA. Please consider rephrasing.
Line 185-187: In particular, ….See comment for Line 154
Line 188 The main tyramine producers are…To our knowledge, Enterococci play a
predominant role in the formation of tyramine.
Pircher A., Bauer F., Paulsen P. (2007). Formation of cadaverine, histamine,
putrescine and tyramine by bacteria isolated from meat, fermented sausages and
cheeses Eur Food Res Technol (2007) 226:225–231
http://dx.doi.org/10.1007/s00217-006-0530-7
Line 195 consider to update the references.
Wüthrich D., Berthoud H., Wechsler D., Eugster E., Irmler S., Bruggmann R. (2017).
The histidine decarboxylase gene cluster of Lactobacillus parabuchneri was gained by
horizontal gene transfer and is mobile within the species. Front. Microbiol., Vol. 8,
Article 218.
https://doi.org/10.3389/fmicb.2017.00218
Line 197: Please consider rephrasing Therefore, the expression of tdc and hdc genes
was checked by qRT-PCR.
Line Consider rephrasing…The strongest increase in tdc expression was found in
R3 cheese after 270 days of ripening (93 fold increase), whereas the other
cheeses showed 61 (R1) and 82 (R2) fold increases of the tdc gene.
Line 202: Consider to replace the word “behavior” by “development”

206-210: Please consider: The formation of the different BAs strongly depends on the
type of microbial contamination in the processed milk. The rather similar
distribution of BA in the three cheeses more likely indicates that the three
different batches of processed milk was contaminated with enterococci (mainly
associated with tyramine) and Enterobacteriaceae (mainly associated with
putrescine, cadaverine).
Line 212: Consider rephrasing and avoid abbreviations in the caption of figures.
Evolution of free amino acids throughout ripening in Pecorino di Farindola
produced with with different rennets. Calf 212 (R1), kid (R2) and pig (R3).
Line 216-217: Consider rephrasing the caption of this figure
Biogenic amines evolution in Pecorino di Farindola manufactured with the
three different rennets. Calf (R1), 216 kid (R2) and pig (R3).
Line 217: Delete the space before the dot ……(R3).
Line 220: Consider rephrasing
……expressed as x-fold increase in comparison with the expression at T0.
Line 223-231 This information should be moved into the introduction section.
Evolution of GABA content in Pecorino di Farindola made with the three
different rennets. Calf (R1), 243 kid (R2) and pig (R3)
Line 243 Consider rephrasing the caption of this figure
Line 251 Proteolysis is not a limiting factor in the formation of BA in ripened cheeses
since there is always a high abundance of FAAs! The most important
prerequisite for the formation of BA are specific contaminations with bacteria
able to decarboxylize specific FAAs. In a cheese environment the ability of
forming BA is an important strain specific advantage (survival of the fittest)
Conclusions:
Line 249 The use of sheep or goat milk is usually associated with higher bacterial
counts and thus with higher contents of BAs. The most reasonable
explanation is that (in comparison to cow milk) a higher number of animals has
to be milked in order to obtain the same quantity of milk. The higher number of
manual operations and the somehow more difficult milking process increase
the risk of contaminations with bacteria associated with the milking equipment
(e.g. Enterococci, Lb. parabuchneri) or the environment.
Line 246-263 Overall, the conclusion section should be carefully revised in order to discuss
the impact of pig rennet and ewes milk in a more comprehensive way.
References:
Line 299: There seems to be a technical problem with the numbering of the references
13-64.
Line 427: There seems to be an other technical problem. Delete “References” 65-72

Author Response

Peer_Review_Foods-582135_v1

Title: Accumulation g-aminobutyric acid and biogenic amines in a traditional raw milk ewe’s cheese,

 General comments:

The present manuscript describes a study on the formation of BAs and GABA in Pecorino di Farindola that is presumably the only cheese variety in the world prepared a traditional preparation of pork rennet.

The experimental design is sound, the type of performed analysis comprehensive. The results are well presented. However, the discussion of the results and mainly the conclusions should be refined. A more comprehensive view of the factors leading to the formation of BA is needed in order to embed the obtained results in the right context.

Without doubt, the manuscript is well written and contains substantial information and therefore should be after major revision.

Specific comments:

Abstract:

Lines 20-23: These results showed that the accumulation of BAs and GABA in Pecorino di Farindola is influenced by ripening time and type of coagulant. Further studies are required to develop starter cultures to reduce BAs content and improve health characteristics of raw milk ewe’s cheeses.

The abstract was revised accordingly (lines 20-23)

Introduction:

Line 42: Please insert some information on the characteristics of pig rennet in comparison to other rennet types (e.g. proteolytic activity, pH optimum, thermo-resistance).

Information about pig rennet has been inserted (lines 44-48)

Line 44: Please consider to rephrase so then the next sentence on NSLAB is better embedded in the context. The microbial composition of the lactic acid bacteria (LAB) present in the starter plays a major role…..On the other hand, adventitious…”

The text was revised accordingly (lines 54-55)

Line 55: Please specify “Biogenic amines, mainly histamine, can lead to …”. The "cheese syndrome" caused by tyramine is rather a side effect of MAO drugs than a food safety issue. The consumption of cheese containing tyramine is unlikely to cause health problems in healthy individuals. According to the EFSA report no adverse health effects were observed after exposure to following BA levels in food (per person per meal): a) 50 mg histamine for healthy individuals, but below detectable limits for those with histamine intolerance; b) 600 mg tyramine for healthy individuals not taking monoamino oxidase inhibitor (MAOI) drugs, but 50 mg for those taking third generation MAOI drugs or 6 mg for those taking classical MAOI drugs; and c) for putrescine and cadaverine, the information was insufficient in that respect.for (EFSA Journal 2011;9(10):2393).

We revised accordingly (lines 67-75)

Line 64: Please add a short listing of the most important positive effects claimed for GABA. … numerous positive effects numerous positive effects on animal and human metabolic disorders such as ……………[16] and therefore gained the attention of the food and pharmaceutical industries.

We revised accordingly (Lines 85-94)

Lines 68-73: Move these sentences to line 34 and consider rephrasing ….cultures [2,3]. In previous studies, the role of pig rennet in the manufacture of Pecorino di Farindola on cheese quality was evaluated. The effect of different types of rennet of animal origin (calf, kid and pig) was compared in terms of physico chemical properties, microbiota, proteolysis, volatile molecule profiles and other characteristics [3,9]. The studies showed that the use of pig makes it possible to distinguish the traditional variant from cheeses made with other coagulants.

We revised accordingly (Lines 48-52)

Materials and Methods:

Line 92: nsert a semicolon (;) …….conditions; enterococci……

We revised accordingly

Line 98: Replace the semicolon by a comma …….[15], Torriani ………..

We revised accordingly

Line 113: Delete the comma after Kőrös …Kőrös et al.

We revised accordingly

Line 116: Please indicate the column type as follows: (Gemini C18, dimensions: 250 ×

4.6 mm, particle size: 5 μm, pore size: 110 Å) (Phenomenex)).

We added this information

Results:

Line 117: “Results and discussion” would be a more appropriate title

We revised accordingly

Line 132: The milking process is important as well. Consider to modify as follows: ….an indicator of the hygienic conditions in milk and cheese production.…..

We revised accordingly (Line 194)

Line 132-133 Delete this part of the sentence …and their absence in the final products reveals good hygienic conditions during the manufacturing process…(Enterobacteriaceae may serve as a process hygiene criterion in cheeses just after manufacture (day 1-14). However, as observed, Enterobacteriaceae disappear rather quickly during cheese ripening. Therefore, the absence of enterobacteriaceae in ripened cheese is not a reliable indicator for good hygienic conditions during the manufacturing process.)

Thank you for this observation, this part of the sentence was deleted

Line 143: Change this sentence as follows: On the other hand, enterococci have often been associated with clinical infections and BAs production, such as tyramine and histamine.

We revised accordingly (Lines 204-205)

 Line 150: It's the other way around. High levels of FAA are rather due to specific species of LAB such as L. helveticus. Consider to delete this sentence

We revised accordingly

Line 154: Reference 35 focuses on g-aminobutyric acid and should be cited together with reference 36 & 37….[35-37]. However,……..

We revised accordingly (line 214)

Line 154: Recently, Lactobacillus parabuchneri has been shown to be the most potent LAB species for the formation of histamine. Consider to rephrase as follows: However, some LAB species have been shown to include strains producing high amounts of BAs such as histamine [new references].

Please consider to include these references in order to update the discussion.

Diaz M., Ladero V., Redruello B., Sanchez-Llana E., del Rio B., Fernandez B., Cruz Martin M., Alvarez M.A. (2016). A PCR-DGGE method for the identification of histamine-producing bacteria in cheese. Food Control 63 (2016) 216e223 http://dx.doi.org/10.1016/j.foodcont.2015.11.035

 Berthoud H. Wüthrich D., Bruggmann R., Wechsler D., Fröhlich-Wyder M.-T., Irmler S. (2017). Development of new methods for the quantitative detection and typing of Lactobacillus parabuchneri in dairy products. International Dairy Journal, 70, 65-71. http://dx.doi.org/10.1016/j.idairyj.2016.10.005

 Ascone P., Maurer J. Haldemann J., Irmler S., Berthoud H., Portmann R., Fröhlich-Wyder M.-T., Wechsler D. (2017). Prevalence and diversity of histamine-forming Lactobacillus parabuchneri strains in raw milk and cheese - a case study. International Dairy Journal, 70, 26-33. http://dx.doi.org/10.1016/j.idairyj.2016.11.012

 Diaz M., del Rio B., Sanchez-Llana E., Ladero V., Redruello B., Fernández M., Cruz Martin M., Alvarez M.A. (2018). Lactobacillus parabuchneri produces histamine in refrigerated cheese at a temperature-dependent rate. International Journal of Food Science and Technology, 53,2342–2348

http://dx.doi.org/10.1111/ijfs.13826

The discussion was updated accordingly

Line 172: As expected, … Long ripened high quality raw milk cheeses such as Parmigiano Reggiano PDO or Emmental PDO usually contain up to 40 g/kg FAA but < 50 mg/kg BA., indicating that there is no correlation between the FAA content and the concentration of BA. Please consider rephrasing.

We thank the reviewer. We rephrased (lines 261-269)

Line 185-187: In particular, ….See comment for Line 154

This sentence was deleted

Line 188 The main tyramine producers are…To our knowledge, Enterococci play a predominant role in the formation of tyramine.

Pircher A., Bauer F., Paulsen P. (2007). Formation of cadaverine, histamine, putrescine and tyramine by bacteria isolated from meat, fermented sausages and cheeses Eur Food Res Technol (2007) 226:225–231 http://dx.doi.org/10.1007/s00217-006-0530-7

We revised accordingly (line 248)

Line 195 consider to update the references.

Wüthrich D., Berthoud H., Wechsler D., Eugster E., Irmler S., Bruggmann R. (2017). The histidine decarboxylase gene cluster of Lactobacillus parabuchneri was gained by horizontal gene transfer and is mobile within the species. Front. Microbiol., Vol. 8, Article 218.

https://doi.org/10.3389/fmicb.2017.00218

We revised accordingly (lines 261-262)

Line 197: Please consider rephrasing Therefore, the expression of tdc and hdc genes was checked by qRT-PCR.

We thank the reviewer. We rephrased (lines 270-271)

 Line Consider rephrasing…The strongest increase in tdc expression was found in R3 cheese after 270 days of ripening (93 fold increase), whereas the other cheeses showed 61 (R1) and 82 (R2) fold increases of the tdc gene.

We thank the reviewer. We rephrased (lines 273-275)

Line 202: Consider to replace the word “behavior” by “development”

We thank the reviewer. We replaced the word

206-210: Please consider: The formation of the different BAs strongly depends on the type of microbial contamination in the processed milk. The rather similar distribution of BA in the three cheeses more likely indicates that the three different batches of processed milk was contaminated with enterococci (mainly associated with tyramine) and Enterobacteriaceae (mainly associated with putrescine, cadaverine).

We thank the reviewer. We rephrased (line 276)

 Line 212: Consider rephrasing and avoid abbreviations in the caption of figures. Evolution of free amino acids throughout ripening in Pecorino di Farindola produced with with different rennets. Calf (R1), kid (R2) and pig (R3).

We thank the reviewer. We rephrased

 Line 216-217: Consider rephrasing the caption of this figure Biogenic amines evolution in Pecorino di Farindola manufactured with the three different rennets. Calf (R1), 216 kid (R2) and pig (R3).

We thank the reviewer. We rephrased

 Line 217: Delete the space before the dot ……(R3).

We revised accordingly

 Line 220: Consider rephrasing……expressed as x-fold increase in comparison with the expression at T0.

We thank the reviewer. We rephrased

Line 223-231 This information should be moved into the introduction section.

We revised accordingly

Line 243 Consider rephrasing the caption of this figure Evolution of GABA content in Pecorino di Farindola made with the three different rennets. Calf (R1), 243 kid (R2) and pig (R3)

We thank the reviewer. We rephrased

Line 251 Proteolysis is not a limiting factor in the formation of BA in ripened cheeses since there is always a high abundance of FAAs! The most important prerequisite for the formation of BA are specific contaminations with bacteria able to decarboxylize specific FAAs. In a cheese environment the ability of forming BA is an important strain specific advantage (survival of the fittest)

This part was deleted

Conclusions:

Line 249 The use of sheep or goat milk is usually associated with higher bacterial counts and thus with higher contents of BAs. The most reasonable explanation is that (in comparison to cow milk) a higher number of animals has to be milked in order to obtain the same quantity of milk. The higher number of

manual operations and the somehow more difficult milking process increase the risk of contaminations with bacteria associated with the milking equipment (e.g. Enterococci, Lb. parabuchneri) or the environment.

 Line 246-263 Overall, the conclusion section should be carefully revised in order to discuss the impact of pig rennet and ewes milk in a more comprehensive way.

This section was rewritten

References:

Line 299: There seems to be a technical problem with the numbering of the references 13-64.

 Line 427: There seems to be an other technical problem. Delete “References” 65-72

We revised all references

Reviewer 2 Report

In this study authors presented a interesting study concerning the influence of different origin rennet on biogenic amines (BAs) and γ-aminobutyric acid (GABA) accumulation in raw milk ewe’s cheeses. 

The study is well designed and presented in most part.

However, I have one crucial issue prior to publication regarding section 2.3 of qPCR analysis. The auhtors report 3 different references and in fact they do not present nothing about how they conducted gene expression assays. This part stronlgy needs further description. For example, how did they obtain bacterial biamass in order to procedd to DNA extraction? Did they use something to stabilise gene expression? Was there an efficient test performed for the reference genes used and what was the values? how was the normalization performed? What was the cycling condition used and melt curve also. How may replicates did they used? Moreover, a table of primers sequences used is totally necesairy to be presented in Methods and Materials section.

Nevertheless, all rest decriptions of experimental processes need to be at least briefly described, reporting just a reference is not the appropriate way of presenting your study.

Last, in conclusion, is suggested the authors do not use any referneces, since they present the outcomes of their spesific study.

References in the manuscript should be appropiately revised according to journal style.

Author Response

In this study authors presented a interesting study concerning the influence of different origin rennet on biogenic amines (BAs) and γ-aminobutyric acid (GABA) accumulation in raw milk ewe’s cheeses.

The study is well designed and presented in most part.

However, I have one crucial issue prior to publication regarding section 2.3 of qPCR analysis. The auhtors report 3 different references and in fact they do not present nothing about how they conducted gene expression assays. This part stronlgy needs further description. For example, how did they obtain bacterial biamass in order to procedd to DNA extraction? Did they use something to stabilise gene expression? Was there an efficient test performed for the reference genes used and what was the values? how was the normalization performed? What was the cycling condition used and melt curve also. How may replicates did they used?

All information required are specified in the text (lines 124-140)

Moreover, a table of primers sequences used is totally necesairy to be presented in Methods and Materials section.

A table was prepared.

Nevertheless, all rest decriptions of experimental processes need to be at least briefly described, reporting just a reference is not the appropriate way of presenting your study.

We better described the methods applied

Last, in conclusion, is suggested the authors do not use any referneces, since they present the outcomes of their spesific study.

Conclusion was revised accordingly

References in the manuscript should be appropiately revised according to journal style.

References were revised

Reviewer 3 Report

It is an interesting, well designed and good scientific quality study. Some minor changes have to be made.

Line 24: please add some other keywords related to the study, such as biogenic amines, hdc gene tdc gene

lines 50, 55 and 159: biogenic amines should be abbreviated. Revise all the manuscript

lines 57-58: Diamine oxidase (DAO) inhibitor drugs can also be responsible of histamine related symtomatology. Please it should be included.

Lines 73-74: Why are not included the microorganisms in the aim?

line 108. Despite the reference of the methods followed to determine biogenic amines it must be included a briefly information about (e.g. the number and type of biogenic amines determined, derivatization reagent, and the High-performance liquid chromatographic instrumentation used as it do in the GABA determination section)

line 121: please add a reference for " it is rich in positive compounds for human health (e.g. GABA)"

lines 125-126: to avoid the personalization (we evaluated) it would be better to write the sentence passively

Line 154: It is true that some strains of LAB are able to produce histamine, but LAB are specially tyramine producers. Include it together with histamine

Lines 183-186: These genera are specially related to the formation of putrescine and cadaverine ? If so, change "amounts of BA" by "amounts of these BA".

Line 186: LAB and staphylococci have also been reported

lines 250-251: this sentence have to be included in the discussion and not in the conclusion section.

from reference 13 the numbers are repeated

references 65-72 are not references of the study and have to be deleted.

Author Response

It is an interesting, well designed and good scientific quality study. Some minor changes have to be made.

Line 24: please add some other keywords related to the study, such as biogenic amines, hdc gene tdc gene

We added new keywords

lines 50, 55 and 159: biogenic amines should be abbreviated. Revise all the manuscript

Biogenic amines was abbreviated as BAs

lines 57-58: Diamine oxidase (DAO) inhibitor drugs can also be responsible of histamine related symtomatology. Please it should be included.

We thank the reviewer. We added this information (lines 78-79)

Lines 73-74: Why are not included the microorganisms in the aim?

We thank the reviewer. We added this information (line 102)

line 108. Despite the reference of the methods followed to determine biogenic amines it must be included a briefly information about (e.g. the number and type of biogenic amines determined, derivatization reagent, and the High-performance liquid chromatographic instrumentation used as it do in the GABA determination section)

We better described the procedure

line 121: please add a reference for " it is rich in positive compounds for human health (e.g. GABA)"

We added the reference (line 184)

lines 125-126: to avoid the personalization (we evaluated) it would be better to write the sentence passively

The sentence was revised

Line 154: It is true that some strains of LAB are able to produce histamine, but LAB are specially tyramine producers. Include it together with histamine

We thank the reviewer. We added this information (Line 215)

Lines 183-186: These genera are specially related to the formation of putrescine and cadaverine ? If so, change "amounts of BA" by "amounts of these BA".

This sentence was deleted

Line 186: LAB and staphylococci have also been reported

The sentence was revised

lines 250-251: this sentence have to be included in the discussion and not in the conclusion section.

The sentence was moved in the discussion (Lines 2248-253)

from reference 13 the numbers are repeated

References were revised

references 65-72 are not references of the study and have to be deleted.

References were revised

Round 2

Reviewer 1 Report

Just two small corrections for the proof reading:

Line 73: Insert a space after "[15]. The" instead of "[15].The"

Line 263: Delete the space "[50,52,53]" instead of "[50, 52,53]"

Reviewer 2 Report

All remarks highlighted by the reviewers have been done.